# Current Trends in Gait Rehabilitation for Stroke Survivors: A Scoping Review of Randomized Controlled Trials

**DOI:** 10.3390/jcm13051358

**Published:** 2024-02-27

**Authors:** Joana Teodoro, Sónia Fernandes, Cidália Castro, Júlio Belo Fernandes

**Affiliations:** 1Department of Nursing, Hospital Garcia de Orta, 2805-267 Almada, Portugal; jfernandes@egasmoniz.edu.pt; 2Nurs* Lab, 2829-511 Almada, Portugal; soniabelo@sapo.pt (S.F.); cidaliamscastro@gmail.com (C.C.); 3Egas Moniz Center for Interdisciplinary Research (CiiEM), Egas Moniz School of Health & Science, 2829-511 Almada, Portugal

**Keywords:** stroke, stroke rehabilitation, rehabilitation, gait, exercise therapy, physical therapy

## Abstract

Background: Stroke stands as a significant global health concern, constituting a leading cause of disability worldwide. Rehabilitation interventions are crucial in aiding the recovery of stroke patients, contributing to an overall enhancement in their quality of life. This scoping review seeks to identify current trends in gait rehabilitation for stroke survivors. Methods: The review followed the methodological framework suggested by Arksey and O’Malley. Electronic databases, such as CINAHL Complete, MEDLINE Complete, and Nursing & Allied Health Collection, were systematically searched in November 2023. Inclusion criteria comprised papers published in either English or Portuguese from 2013 to 2023. Results: From the initial search, a total of 837 papers were identified; twenty-one papers were incorporated into this review. Thirteen distinct categories of gait rehabilitation interventions were identified, encompassing diverse approaches. These categories comprise conventional rehabilitation exercises, traditional gait training with integrated technology, and gait training supported by modern technologies. Conclusions: Although traditional rehabilitation exercises have historically proven effective in aiding stroke survivors, a recent trend has emerged, emphasizing the development and integration of innovative therapeutic approaches that harness modern technologies.

## 1. Introduction

Stroke stands as a substantial global health issue, holding the position of second-leading cause of mortality and the third-leading cause of disability on a worldwide scale.

The burden of stroke is staggering, with six and a half million deaths recorded in 2019, along with over 12.2 million new stroke cases and a prevalence of 101 million stroke survivors [1]. These data emphasize the considerable burden of stroke on public health, encompassing mortality and disability, solidifying its status as one of our era’s most prevalent and impactful diseases [2].

Functional changes are frequently observed among stroke survivors [3]. Patients commonly experience cognitive and motor impairments that affect balance, coordination, proprioception, muscle tone, muscle strength, and gait, making them prevalent sequelae of stroke [3,4].

A prior study revealed that up to 83% of stroke survivors experience balance impairment [5]. Motor deficits resulting from a stroke can significantly impact an individual’s functional independence. Challenges in balance, coordination, muscle strength, and gait often lead to difficulties in performing basic activities of daily living, such as walking, dressing, and eating. These limitations also impact stroke survivors’ abilities to engage in social interactions, pursue leisure activities, and return to work after a stroke [6,7], leaving individuals dependent on others for support and assistance, impeding their sense of autonomy and self-reliance. Moreover, balance impairments significantly contribute to adverse outcomes following a stroke, including an increased risk of falls [8,9], fall-related injuries such as fractures [10], fear of falling [11], and even mortality [12]. Furthermore, a previous investigation established a correlation between balance deficits in the acute stage of stroke and subsequent cognitive impairment one year post-stroke [13].

The repercussions of a stroke are severe, resulting in various physical, cognitive, and emotional challenges. The impact extends beyond individual health, significantly influencing overall wellbeing and the capacity for an independent and fulfilling life [14,15]. Effectively addressing these challenges through gait rehabilitation enhances their quality of life and facilitates their reintegration into society [16].

Rehabilitation interventions are crucial in assisting stroke survivors in recovering motor function and enhancing their overall quality of life. By improving mobility, motor skills, and functional abilities, stroke survivors can achieve greater independence, leading to a more fulfilling and satisfying life post-stroke [17,18].

The management of stroke demands specialized care from healthcare professionals in both hospital and community settings. Survivors often require comprehensive rehabilitation and support to regain functional abilities and adapt to a new reality [19,20]. Families of stroke survivors also face significant adjustments as they navigate unfamiliar challenges and seek ways to provide necessary care and assistance [21,22].

Gait rehabilitation is crucial in stroke survivors’ comprehensive recovery and functional independence [23]. As stroke remains a leading cause of disability worldwide, exploring current trends in gait rehabilitation for this population is of utmost importance. 

Developing effective rehabilitation programs after stroke is imperative to address this pressing health issue. By identifying and understanding the impact of rehabilitation interventions on gait recovery, we can enhance the rehabilitation process for stroke survivors, ultimately improving their functional independence and overall quality of life. Therefore, this review seeks to identify current trends in gait rehabilitation for stroke survivors.

## 2. Methods

The current scoping review adheres to the methodological framework outlined by Arksey and O’Malley [24], which encompasses five distinct stages. Furthermore, to strengthen the scoping study methodology and ensure consistent reporting, we incorporated recommendations outlined in the Preferred Reporting Items for Systematic Reviews and Meta-Analyses extension for Scoping Reviews (PRISMA-ScR) Checklist [25].

### 2.1. Stage 1: Identifying Research Questions

The initial stage involves formulating a precise research question that guides the review. The research question framed for this review was: What are the current trends in gait rehabilitation for stroke survivors?

### 2.2. Stage 2: Identifying Relevant Studies

On 10 November 2023, the initial search occurred across three databases on the EBSCOhost research platform: CINAHL, MEDLINE, and Nursing & Allied Health Collection.

A search strategy was planned using the Population, Concept, Context (PCC) framework, incorporating the MeSH terms.

The search strategy employed in CINAHL Complete was as follows:S1: “Stroke”S2: “Rehabilitation”S3: “Gait”

Combined search: S1 AND S2 AND S3.

The inclusion/exclusion criteria are outlined in Table 1.

The search was limited to articles published between 2013 and 2023. 

Two researchers (J.T. and J.B.F.) independently performed the search, stages 2 to 4. 

Given constraints in translation resources, the review excluded papers published in languages other than English or Portuguese.

### 2.3. Stage 3: Study Selection

All citations were imported into Rayyan—an AI-powered tool for Systematic Literature Reviews, where duplicate citations were identified and eliminated. Subsequently, the citations underwent title and abstract relevance screening. We examined the full text of pertinent papers, including those that met the study criteria for this review.

In cases of uncertainty regarding whether an article fits the review criteria, it proceeded to the subsequent phase. Reviewers held regular meetings throughout the screening to resolve conflicts and address any uncertainties about selecting papers. A third reviewer (S.F.) made the final and conclusive decision if any disagreements occurred.

### 2.4. Stage 4: Data Charting

A systematic approach was used to retrieve data. Two reviewers, J.T. and J.B.F., extracted data using a customized instrument to collect relevant information addressing the research question. The extracted data encompassed general details (author’s name, publication year, title, and country), methodological specifics (study design and aim), and results (interventions employed by health professionals for gait rehabilitation in stroke survivors). Subsequently, all authors thoroughly reviewed and discussed the final data extraction chart.

### 2.5. Stage 5: Collating, Summarizing, and Reporting the Results

Researchers prepared a PRISMA flow diagram illustrating the study identification, screening, and selection process (Figure 1). To organize and summarize the data, a data-driven thematic analysis was implemented, adhering to the guidelines established by Braun et al. [26]. Two researchers, J.T. and S.F., independently carried out the data review, employing manual coding and analysis to identify recurrent themes in the data.

## 3. Results

The initial search resulted in 837 potentially relevant articles. After removing duplicates and conducting relevance screening based on title and abstract, 517 articles were excluded. Subsequently, 21 articles met the eligibility criteria, full articles were reviewed, and were considered suitable for the review. Figure 1 illustrates the flow chart depicting the screening process.

The 21 included studies spanned the period from 2013 to 2023. Among the 21 studies, 10 were conducted in South Korea [27,28,29,30,31,32,33,34,35,36], 1 in North Korea [37], 1 in Poland [38], 1 in Serbia [39], 1 in Pakistan [40], 1 in United States of America [41], 1 in Italy [42], 1 in Turkey [43], 1 in Egypt [44], 1 in The Netherlands [45], 1 in Taiwan [46] and another in China [47] (Table 2).

Data analysis uncovered a range of interventions to enhance gait in stroke survivors. The interventions were classified into 13 groups, and each of these categories is explained below.

### 3.1. Backwards Walking

In the study by Kim et al. [34], participants engaged in a program that combined backward treadmill walking with a gradual reduction in weight support. The systematic reduction in weight support occurred over four weeks, starting with a 40% reduction in the first week and a 10% reduction in subsequent weeks. Participants maintained an average speed ranging from 0.08 to 0.22 m/s, with a 0.1 km/h speed increment introduced at each session. To assist during training, two physical therapists played crucial roles—one positioned behind the subject for weight support and movement guidance and another set by the paretic leg to aid in motor control throughout the gait cycle.

In contrast, the control group underwent conventional forward treadmill training with matched duration and frequency. Results showed that the intervention group exhibited improvements in all dependent variables by week four compared to the control group. In the Munari et al. [42] study, both groups started treadmill training at 60% of their baseline over-ground self-selected speed, determined during gait analysis, with a 1% incline. The findings showed significant improvement in the 10 m Walking Test and stabilometry assessment post-treatment. Remarkably, superior enhancements in both gait and balance were noted following backward treadmill training compared to forward treadmill training.

### 3.2. Tai Chi 

Yu et al. [47] conducted a study in which participants wore a harness connected to an overhead suspension system, supporting a specific percentage of their body weight. The intervention was based on seven steps from the 24-form simplified Tai Chi recommended by the State Sports General Administration of China, encompassing forward steps, backward steps, shuffle steps, empty steps, lunge steps, single-leg support, and turning around. These steps constitute the foundational elements of Tai Chi movements. The training primarily focused on lower limb movements, emphasizing endurance and weight shifts, rather than upper limb actions. The training program followed a gradual progression corresponding to a 10% reduction in body weight support.

In contrast, the control group underwent conventional rehabilitation training. The findings reveal significant differences between groups in scores related to directional control during the limits-of-stability test. Additionally, the Tai Chi group exhibited superior scores in gait cycle time, step length, step velocity, and range of motion of the joints compared to the control group.

### 3.3. Dual-Task Training

Saleh et al. [44] investigated the impact of aquatic versus land motor dual-task training. The training involved walking while holding a ball and a cup of water and standing on a balance board with a moving cup in various walking conditions (forward, sideways, and backward). In the water-based exercises, participants performed these tasks in a large swimming pool, while the land group executed the same sequence on solid ground. Both groups showed significant improvements in all outcome variables post-treatment, with the water-based training group demonstrating superior results in overall stability, anteroposterior stability, mediolateral stability, walking speed, step length of affected and non-affected limbs, and time of support on the affected limb compared to the land-based group.

Hong et al. [30] employed dual-task training involving familiar traffic signals. The intervention incorporated scaffolds with marked starting and target points, monitors displaying visual cues for the cognitive task, and elastic bands controlling resistance amounts and difficulties. For the cognitive task training group, the task involved standing and moving the less-affected lower extremity in three hip flexion directions based on red and green cues resembling traffic lights on the monitor. Three traffic signals represented the directions, and colors and locations changed randomly. Cognitive balance training progressed in difficulty: without elastic bands, an elastic band, and a differently colored elastic band for increased resistance. Elastic bands were positioned around the ankle of the less-affected side.

The general task training group performed a similar lower extremity movement without the cognitive task. Significant differences were observed in both groups after the intervention. The cognitive task training group significantly improved all outcome scores after the intervention.

In the study by Iqbal et al. [40], participants in the dual-task training group walked backward, sideways, and forward while holding a 100 g sandbag. Additionally, they performed tasks such as picking up plastic cups in front of their feet. The control group received conventional physiotherapy, encompassing stretching, strengthening exercises, and gait training. Post-treatment assessments revealed a significant enhancement in the 10 m walk, cadence, step length, stride, and cycle time within the dual-task training group.

### 3.4. Action Observation Training and Motor Imagery Training

Kim and Lee [33] compared the effects of action observation and motor imagery training on stroke survivors. In the action observation training group, participants watched a 20 min task video. This was followed by 10 min of physical training with a therapist based on the video. The video featured adult models performing motions relevant to each participant’s hemiplegia. The training program, divided into four stages, focused on trunk stability, mobility, sit-to-stand, weight shifting, and gait improvement. Each stage’s video was viewed weekly to reduce individual deviations based on the hemiplegia side. In the motor imagery training group, participants spent 20 min on motor imagery through a computer speaker and 10 min of physical training based on the imagery program. The motor imagery program content mirrored the action observation training program. Participants trained on each stage’s content for one week. All participants received neurodevelopmental therapy focusing on trunk and lower extremity movements, sit-to-stand, and gait patterns on level surfaces and stairs. The action observation training group significantly improved gait speed, cadence, and single limb support compared to the physical training group. However, no significant differences were observed in any of the outcome measures.

### 3.5. Visual Biofeedback

Druzbicki et al. [38] assessed the impact of gait training using a treadmill, comparing outcomes with and without visual biofeedback. The intervention group utilized a Gait Trainer 2 treadmill featuring real-time visualization of foot placement and the designated foot positioning area. The intervention group engaged in gait training on the treadmill with visual biofeedback, which included step length, foot placement location, and an acoustic signal-confirming task execution accuracy. In contrast, the control group underwent treadmill training without biofeedback. The results indicated that the intervention group yielded superior outcomes, particularly in enhancing the gait cycle, duration of gait phases, and speed of the swing phase. The results indicated that the intervention group achieved superior outcomes, especially in improving the gait cycle, duration of gait phases, and speed of the swing phase.

### 3.6. Vibration Therapy

Choi et al. [29] explored the impact of whole-body vibration and treadmill training on walking performance using a side-alternating vibrator (Galileo 2000, Novotec, Nettetal, Germany, 2011). Whole-body vibration involved a maximum frequency of 30 Hz and an amplitude of 3 mm, lasting 45 s. Participants stood on the vibration platform with their feet parallel to the axis, lightly holding a support bar. Each session comprised six exercises, each lasting 45 s, including weight shifts, squats, anteroposterior weight shifts, forward lunges, one-leg standing, and deep squats. A 1 min break separated each exercise. The control group performed the identical exercise program minus vibration therapy. The vibration therapy group significantly improved walking speed, cadence, and temporal parameters. In contrast, the control group exhibited improvement solely in walking speed.

### 3.7. Functional Electrical Stimulation

Hwang et al. [31] used treadmill gait training combined with the WalkAide system, incorporating an inclination sensor for functional electrical stimulation. The WalkAide stimulator, a compact electronic device, was affixed to the common peroneal and anterior tibial nerves, delivering electrical stimulation based on knee flexion angles during walking. The WalkAnalyst program determined the optimal stimulus intensity, followed by treadmill gait training guided by the inclination sensor to facilitate ankle dorsiflexion. The findings suggested that gait training on a treadmill with functional electrical stimulation effectively enhanced balance, gait, and anterior tibial structure.

In the study by Dujović et al. [39], the Functional Electrical Stimulation group, electrical stimulation targeted the tibial nerve in the pre-swing phase and the common peroneal nerve in the swing phase, inducing ankle plantar flexion and dorsiflexion, respectively. The Functional Electrical included a stimulation unit, demultiplexer, clothing with integrated multi-pad electrodes, wireless inertial sensors, and a tablet PC with a dedicated application. The multi-pad electrode garment strategically placed around the knee stimulated the common peroneal and tibial nerves. The stimulator unit delivered a customized pulse train to the multi-pad electrode, with parameters set during calibration (frequency of 40 Hz and pulse width of 400 ms). Results revealed that combining functional electrical stimulation with conventional rehabilitation was more effective in improving walking speed, lower limb mobility, balance, and daily activities compared to conventional rehabilitation alone.

Yang et al. [46] utilized electromyographic-triggered neuromuscular electrical stimulation with two surface electrodes targeting the anterior or medial gastrocnemius of the tibialis. Electrodes were placed at motor points, and the stimulation frequency was set at 50 Hz with a pulse width of 0.2 ms, using a biphasic square wave. Each session lasted 20 min, with a stimulation cycle of 5:15. Participants actively performed dorsiflexion or plantar flexion during sessions based on electromyographic signals of maximal voluntary contraction. Training intensity ranged from 50 mV to 0 mV to ensure a comfortable full range of motion. After the 20 min stimulation, participants engaged in 15 min of ambulation training with verbal cues emphasizing specific ankle movements. The study concluded that neuromuscular electrical stimulation applied to ankle dorsiflexion and ambulation training effectively strengthened muscles, reduced spasticity, and improved ankle control during push-off and gait performance. Similarly, using neuromuscular electrical stimulation to ankle plantar flexors with ambulation training positively affected gait temporal symmetry in chronic stroke survivors with insufficient ankle control.

### 3.8. Rhythmic Auditory Stimulation

Cha et al. [28] used rhythmic auditory stimulation in group sessions, incorporating personalized music tapes and a metronome tailored to individual musical preferences. This synchronization aimed to improve rhythmic perception and align with the walking patterns of participants.

The training started with participants becoming familiar with the music’s rhythm and coordinating hand and foot movements to the beat. Participants were instructed to walk while synchronizing their movements with the music and metronome to facilitate a seamless transition to coordinated steps. Throughout the sessions, participants engaged in intensive gait exercises enriched by rhythmic auditory stimulation. As the training sessions advanced, the reliance on rhythmic stimulation was systematically reduced, prompting participants to practice intensive gait training independently. The researchers ultimately determined that this all-encompassing approach to intensive gait training, incorporating rhythmic auditory stimulation, significantly enhanced balance and gait performance among stroke survivors.

### 3.9. Gait Assistive Devices

Kang et al. [32] employed a Weight Support Feedback Cane and a smartphone application that quantitatively measured cane dependence during walking. The Weight Support Feedback Cane facilitated real-time assessment of cane dependence, displaying the information on the cable and the smartphone app. Participants determined a weight-bearing rate based on cane dependence, ranging from 60% to 30%, and made weekly adjustments. An audible signal alerted participants if the loaded weight exceeded the preset rate, continuing until it fell below the limit. Based on the baseline measurement of cane dependence, the weight-bearing rate progressively decreased by 10% weekly, from 60% to 30%. The gait success rate, documented by the smartphone app, influenced the reduction in the weight-bearing rate for the subsequent week.

Results demonstrated that both cane gait training methods (traditional vs. Weight Support Feedback Cane) significantly enhanced lower limb muscle activity and walking ability. The Weight Support Feedback Cane group exhibited more benefits than conventional cane gait training.

Lee et al. [35] explored the impact of a Wearable Tube-Assisted Walking Device on gait parameters in stroke survivors. The device comprises a pelvic belt, an elastic tube, and a conventional elastic orthosis created with an open sock and two strips of elastic material. Elastic tubing was employed to provide elasticity assistance, with patients using tubes half the length of their leg for the test. The application process involves placing the sock on the affected limb, attaching long and short straps on opposite sides, passing over the feet, and securing the pelvic belt. Within this setup, one tube end is affixed to a hole in the conventional elastic orthosis, while the other side is hooked into the pelvic girdle ring. The tubing generates superior traction force, aiding knee flexion during push-off and swing phases of gait by converting stored potential energy into kinetic energy. Results indicated that the Wearable Tube-Assisted Walking Device effectively enhanced gait speed, cadence, and stride length.

### 3.10. Fresnel Prism Glasses

Ha and Sung [37] explored the efficacy of prismatic Fresnel glasses in balance and gait training for stroke survivors on an electronic treadmill. The Fresnel prism, set at a 15-diopter deflection angle and tailored to individual patients, was applied contralaterally to the hemiplegia. This induced an adaptive effect, translating spatial information into retinal perceptual coordination. Despite no impact on visual perception, the prism significantly enhanced balance and walking ability in stroke patients with hemiplegia and no visual impairment. The use of Fresnel prismatic glasses, even in the absence of visual issues, positively influenced balance and gait training for stroke survivors, underscoring the therapeutic potential of these glasses in improving spatial movement and motor tasks.

### 3.11. 3D Spine Balance System

Moon and Kim [36] explored the impact of the newly developed three-dimensional (3D) Spine Balance system on stroke survivors. Subjects underwent 30 min of central nervous system development therapy, with the experimental group incorporating additional spinal stability exercises using the Spine Balance 3D system. The Spine Balance 3D system, equipped with diagnostic, exercise, and game modes, was utilized in exercise mode for the study. In this mode, the tilt angle was adjustable from 5° to 60° in eight directions. During training, the system tilted the entire body of subjects while maintaining a straight, neutral position, applying gravity to the torso for stability within the body’s line of gravity. The pelvis, sacrum, and femur were secured, and participants crossed their arms over their chests under the inspector’s control to prevent torso compensation against gravity. The slope gradually increased, incorporating different tilt angles and torque levels for varied training intensities. Results show that the Spine Balance 3D system enhanced trunk muscle strength and walking ability in chronic stroke patients more effectively than conventional training.

### 3.12. Augmented Reality-Based Training

Timmermans et al. [45] utilized augmented reality training to enhance walking adaptability. This involved the projection of gait-dependent contextual cues onto the treadmill surface to prompt step adjustments. The training regimen encompassed a variety of exercises, including navigating visual obstacles, adjusting foot positioning in response to regular or irregular sequences of stepping targets (goal-directed stepping), managing gait acceleration and deceleration within a moving projected walking area on the treadmill, tandem walking, and an interactive walking-adaptability game. In contrast, the standard overground program aimed to reduce post-stroke falls by incorporating walking-adaptability exercises. This program included an obstacle course with exercises on obstacle avoidance, foot positioning on uneven surfaces, tandem walking, and slalom walking. The results revealed no significant group differences for the primary outcome measure. Augmented reality training resulted in twice as many steps per session, with equal duration compared to the standard overground program.

### 3.13. Robot-Assisted Gait Training

Kelley et al. [41] examined the efficacy of robotic-assisted body weight-supported treadmill training utilizing the Lokomat^®^. In the intervention group, participants were supported by a harness connected to a body weight support system that was adjusted based on individual strength and conditioning levels. The weight was gradually decreased as tolerated. The Lokomat^®^ facilitated sagittal plane assistance for hip and knee joint movements, mimicking a symmetrical reciprocal gait. Participants received visual feedback on their walking pattern through a mirror and a computer display illustrating bilateral hip and knee motions. The walking speed was gradually increased from 0.42 m per second (m/s) to a maximum of 0.89 m/s as tolerated. Guidance force, denoting the assistance provided by the robot-driven gait orthosis for moving the legs through prescribed sagittal plane motions, started at 100% for both legs and was subsequently reduced as participants demonstrated proficiency in executing the movements. This study detected intragroup variations in the Fugl-Meyer Lower Extremity Motor score and Barthel Index from baseline to post-intervention and baseline to the 3-month follow-up. Nevertheless, no differences were observed between the two groups.

Bang and Shin [27] conducted a comparative analysis to assess the impact of robot-assisted gait training (utilizing Lokomat^®^) versus treadmill gait training, following a protocol like the Kelley et al. [41] study. The findings demonstrated differences between groups, with the intervention group exhibiting significantly higher gait speed, cadence, step length, and activities-specific balance confidence score than the control group. The robot-assisted gait training group also significantly reduced the double limb support period. Mustafaoglu et al. [43] adopted a protocol similar to previous studies involving the application of Lokomat^®^. However, they organized participants into three groups: a conventional training group, a robot-assisted gait training group, and a combined training group that received both conventional and robot-assisted gait training. Their findings indicate the mean change in all primary and secondary outcomes, except the Fast 10 m Walk Test. In the subgroup analysis, the combined training group demonstrated significant improvements in the Barthel Index, Stroke-Specific Quality of Life Scale, 6 min Walk Test, and Stair Climbing Test compared to the other two groups.

## 4. Discussion

The scoping review delves into the evolving landscape of gait training for stroke patients, shedding light on emerging trends extending beyond the conventional rehabilitation exercises traditionally employed in stroke therapy (Figure 2). 

By synthesizing various studies and identifying patterns and trends in gait rehabilitation, this review provides insights into the rationale behind adopting novel interventions, highlighting their advantages over traditional approaches. Moreover, the review offers a holistic view of the current state of gait training, emphasizing the importance of incorporating innovative strategies to enhance patient outcomes and quality of life.

Furthermore, the review is a practical guide for professionals considering incorporating these interventions into their practice. By identifying and describing interventions, the review enables professionals to make informed decisions about integrating new approaches into their service. This state of play enhances the knowledge base within the field and empowers practitioners to adapt and evolve their treatment protocols to meet stroke survivors’ needs better.

While recognizing the well-established effectiveness of conventional rehabilitation exercises for stroke survivors, this review identifies several trends in gait training that surpass traditional approaches: (1) The review highlights a shift towards developing and adopting novel interventions beyond traditional rehabilitation exercises. These interventions include Tai Chi [47], dual-task training [30,40,44], action observation training [33], visual biofeedback [38], vibration therapy [29], and robot-assisted gait training [27,41,43]; (2) There is a trend towards adopting a holistic approach to gait rehabilitation, which involves addressing physical, cognitive, and sensory aspects of gait recovery. Interventions like dual-task [30,40,44] and action observation training [33] incorporate cognitive components, while others, such as vibration therapy [29], target sensory stimulation; (3) There is an emphasis on individualized treatment plans tailored to the specific needs and capabilities of stroke survivors. Interventions like robot-assisted gait training [27,41,43] and wearable assistive devices [32,35] allow customized adjustments and feedback based on individual progress and abilities; and (4) Many emerging interventions leverage technological advancements to enhance gait training outcomes. Examples include wearable sensors, robotic devices, and biofeedback systems to provide real-time feedback and improve patient engagement during rehabilitation.

Several studies have explored a range of gait rehabilitation approaches, including traditional methods with slight modifications such as backward walking through treadmill training with body weight support [34] and straightforward treadmill training [42]. Additionally, the innovative practice of Tai Chi with body weight support has proven beneficial for stroke survivors, leading to significant improvements in directional control and gait patterns [47]. Dual-task training has emerged as a promising avenue, with studies investigating its application through motor tasks in both aquatic and terrestrial environments [44], exclusively in terrestrial settings [30,40]. These varied approaches underscore the importance of introducing exercise variations into traditional practices to optimize rehabilitation outcomes. Action observation and motor imagery training [33] represent another innovative dimension in gait rehabilitation after a stroke. These methods involve integrating conventional exercises with video-based demonstrations and mental visualization, emphasizing the strategic role of instructional materials in optimizing the effectiveness of rehabilitation interventions.

Blending complementary interventions with conventional rehabilitation exercises demonstrates considerable potential for enhancing treatment outcomes in stroke survivors [48].

Integrating complementary interventions alongside conventional rehabilitation exercises holds significant promise for enhancing treatment outcomes in stroke survivors [48]. By incorporating diverse approaches, healthcare professionals aim to tailor rehabilitation experiences for each patient, leading to improved outcomes [49,50]. These novel strategies target the physical aspects of gait and emphasize motivating and engaging patients, enhancing adherence to rehabilitation plans [23,51]. Incorporating interactive therapies can make rehabilitation more enjoyable, potentially improving compliance with prescribed exercises [52,53].

Innovative therapeutic options in stroke rehabilitation, such as augmented reality-based training, robotic-assisted therapy, rhythmic auditory stimulation, and functional electrical stimulation, tap into survivors’ various physical and cognitive abilities. This multi-faceted approach to gait training may promote neural plasticity, encouraging the brain to adapt and reorganize in response to the specific demands of each intervention. Promoting neural plasticity is crucial for stroke survivors, as it enhances the brain’s ability to form new connections, compensate for damaged areas, and support functional recovery [54,55,56]. Therefore, incorporating these innovative therapies represents a promising strategy to optimize rehabilitation outcomes and improve overall functional abilities in survivors. For instance, visual biofeedback [38] in gait rehabilitation demonstrates substantial improvements when combined with treadmill training. Real-time visualization of foot placement and acoustic signals provides an interactive and adaptive training environment. Body vibration Therapy [29] introduces an additional sensory element to treadmill training, offering an innovative approach to optimizing gait performance. Functional electrical stimulation [31,39] incorporates adaptive stimulation, targeting specific nerves during the gait cycle. Intervention based on rhythmic auditory stimulation [28] introduces music as a synchronizing method to improve rhythmic coordination in gait, highlighting the importance of auditory perception in rehabilitation and complementing visual stimuli.

The evolution of rehabilitation highlights a notable shift towards interventions centered exclusively on modern technology, exemplified by innovations such as the weight-bearing feedback cane [32]. This approach underscores the ingenuity in real-time monitoring of weight-bearing during gait, incorporating sound feedback and customizable weight support. Prismatic Fresnel glasses [37] contribute a personalized dimension to visual intervention for hemiplegic patients without visual impairment, offering individually adjusted solutions. The 3D spine balance system [36] utilizes a three-dimensional device for stability exercises, showcasing targeted and specific technological applications for post-stroke patients. Augmented reality-based Rehabilitation [45] represents another stride in leveraging technology, employing interactive projection on the treadmill to generate dynamic and adaptive responses, enhancing the efficiency of gait recovery. Robot-assisted gait training, particularly with Lokomat^®^ [27,41,43], is an innovative and valuable approach, providing specific benefits in personalization, control, and visual feedback. Combining conventional training with Lokomat^®^ training [43] can amplify health gains and improve the quality of life for stroke survivors. This integration showcases the evolving landscape of rehabilitation, emphasizing the strategic incorporation of advanced technologies to enhance the precision and efficacy of interventions.

A notable trend identified in this review is using robotic devices to aid gait training. Robotic equipment offers several advantages, including increased patient independence to engage in more repetitive and customized training sessions. These devices provide precise control and monitoring of movements, allowing therapists to tailor rehabilitation programs to individual needs and track progress effectively [57]. Moreover, incorporating robotic devices has the potential to program rehabilitation exercises for patients to perform multiple daily sessions. This scheduling flexibility provides stroke survivors with increased opportunities for practice and eases the burden on healthcare professionals by reducing the need for constant supervision during each session [58].

The availability of robotic-assisted gait training presents a promising avenue for stroke rehabilitation, as it addresses the demand for intensive, repetitive, and task-specific exercises while optimizing the use of healthcare resources [59,60,61]. However, despite the promising advantages of robotic-assisted gait training in stroke rehabilitation, these technological devices are not widely utilized in current rehabilitation practices. One significant barrier to their widespread adoption is the substantial financial investment required [62]. Robotic devices often come with high costs that many rehabilitation centers and hospitals may find challenging to afford within their budget constraints.

Furthermore, these devices usually necessitate specialized training and supervision from healthcare professionals, making them challenging to use independently by stroke survivors in their homes [63]. Moreover, the space and setup required for these devices may not be feasible in typical home environments, further hindering their application outside clinical settings [64,65]. As this technology continues to evolve, assessing its effectiveness, cost-effectiveness, and long-term impact on functional outcomes is essential to effectively guide its integration into clinical practice. Long-term follow-up studies and cost-effectiveness analysis are particularly crucial in this regard. By conducting further research and evaluation, healthcare professionals can better understand the potential benefits and limitations of robotic-assisted gait training, thus optimizing its implementation and maximizing its positive impact on stroke rehabilitation outcomes.

Innovative rehabilitation interventions have the potential to motivate stroke survivors to engage in rehabilitation programs by offering fresh and varied approaches compared to conventional training methods [66,67]. These interventions provide stimulating challenges that encourage active rehabilitation involvement, fostering patient motivation and engagement. Additionally, they offer immediate and visible feedback on patient performance, helping individuals understand their progress and feel motivated to continue with the rehabilitation program.

The integration of complementary interventions alongside conventional exercises further enhances treatment outcomes for stroke survivors. By combining traditional rehabilitation exercises with innovative approaches, patients benefit from a more comprehensive and holistic approach to rehabilitation. These complementary interventions target different aspects of recovery, addressing cognitive, sensory, and motor functions. As a result, stroke survivors experience enhanced engagement, improved functional outcomes, and a higher likelihood of sustained progress throughout their rehabilitation [63,68,69].

### Strengths and Limitations

A notable strength of this review lies in its dedicated focus on randomized controlled trials. Our findings offer valuable insights into several approaches for gait rehabilitation, meticulously studied in experimental settings. The comprehensive overview of these interventions fosters a deep understanding of their techniques and methodologies, providing health professionals with the knowledge to replicate them confidently. This emphasis on evidence-based practices enhances the applicability and effectiveness of gait rehabilitation strategies for stroke survivors.

One potential limitation of this review is its inclusion of studies published exclusively in English or Portuguese. This approach introduces language bias and may overlook valuable research in other languages. Consequently, the comprehensiveness of the review could be limited by not considering a broader range of studies from diverse linguistic sources. Furthermore, despite analyzing many results, the restriction of the search to the EBSCO-host research platform is one of the major limitations that may have led to the exclusion of relevant studies from other sources such as Cochrane Central Register of Controlled Trials (CENTRAL) and Embase. Acknowledging these limitations is crucial to ensure transparency and a balanced interpretation of the review’s findings.

## 5. Conclusions

While conventional rehabilitation exercises have long been effective in treating stroke survivors, recent years have seen an increase in the development and integration of new therapeutic approaches involving modern technologies. These complementary interventions improve gait recovery by providing innovative and engaging options for stroke survivors. The complementarity of conventional and technological exercises highlights the importance of an integrated approach in searching for more comprehensive results adapted to each patient’s needs.

However, more research is needed to determine these new interventions’ effectiveness and long-term benefits, which may guide the future direction of stroke rehabilitation strategies.

## Figures and Tables

**Figure 1 jcm-13-01358-f001:**
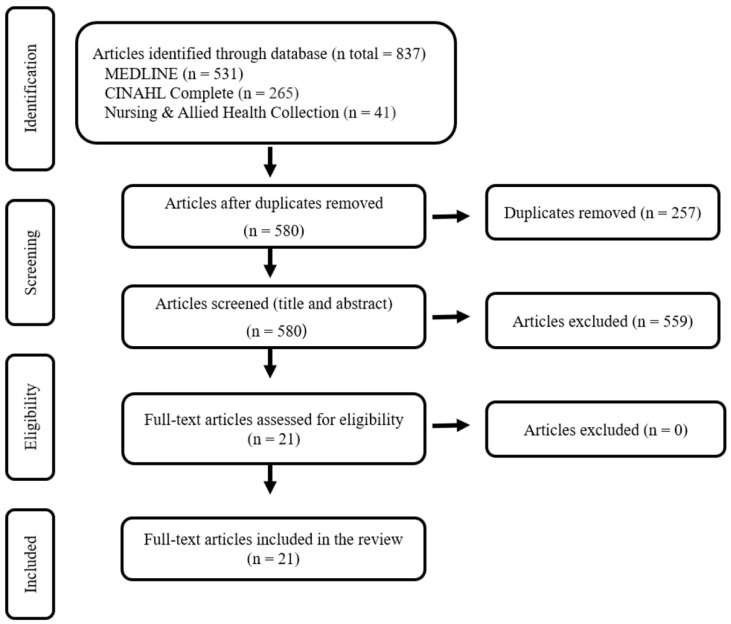
PRISMA flow chart for study selection.

**Figure 2 jcm-13-01358-f002:**
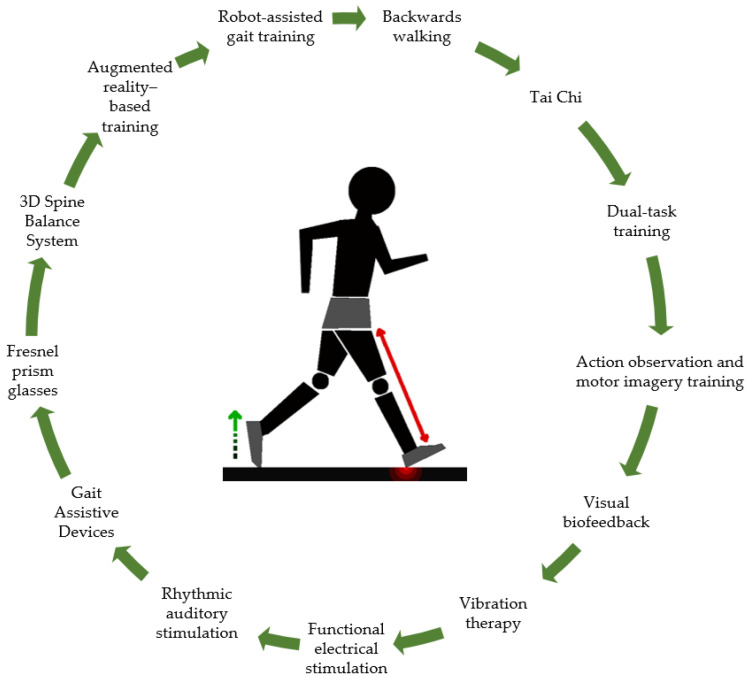
Categories of gait rehabilitation interventions.

**Table 1 jcm-13-01358-t001:** Inclusion/exclusion criteria.

Parameter	Inclusion Criteria	Exclusion Criteria
Population	Stroke survivors; Adults ≥ 18 years old.	Other health conditions besides stroke; Participants < 18 years old.
Concept	Studies that explore interventions for promoting gait.	Studies that do not explore interventions for promoting gait.
Context	Studies conducted in rehabilitation settings.	Studies conducted in non-rehabilitation settings.
Study design	Randomized controlled trials focusing on interventions that promote gait in stroke survivors.	Other type of studies.

**Table 2 jcm-13-01358-t002:** Data extraction and synthesis.

Author/Year/Country	Study Design/Study Aim	Intervention
Bang and Shin [27] (2016) South Korea	Randomized controlled trial To compare the effects of robot-assisted gait training versus treadmill on spatiotemporal gait parameters, balance, and activities-specific balance confidence with stroke patients.	Robot-assisted gait training
Cha et al. [28] (2014) South Korea	Randomized controlled trial To investigate the effect of intensive gait training with rhythmic auditory stimulation on postural control and gait performance in individuals with chronic hemiparetic stroke.	Rhythmic auditory stimulation
Choi et al. [29] (2017) South Korea	Randomized controlled trial To investigate the effect of whole-body vibration combined with treadmill training on walking performance in patients with chronic stroke.	Vibration therapy
Druzbicki et al. [38] (2015) Poland	Randomized controlled trial To evaluate the effects of gait training using a treadmill with and without visual biofeedback in patients late after stroke and to compare both training methods.	Visual biofeedback
Dujović et al. [39] (2017) Serbia	Single-blind randomized trial To evaluate the efficacy of an additional novel FES system to conventional therapy in facilitating motor recovery in the lower extremities and improving walking ability after stroke.	Functional electrical stimulation
Ha and Sung [37] (2020) North Korea	Randomized controlled trial To investigate the effect of Fresnel prism glasses on balance and gait in stroke patients with hemiplegia.	Fresnel prism glasses
Hong et al. [30] (2020) South Korea	Randomized controlled trial To determine whether cognitive task training improves walking and balancing abilities for stroke survivors.	Dual-task training
Hwang et al. [31] (2015) South Korea	Randomized controlled trial To investigate the effects of treadmill training with tilt sensor FES on the balance, gait, and muscle architecture of the tibialis anterior in stroke survivors.	Functional electrical stimulation
Iqbal et al. [40] (2020) Pakistan	Randomized controlled trial To compare the effectiveness of dual task-specific training and conventional physical therapy in the ambulation of patients with chronic stroke.	Dual-task training
Kang et al. [32] (2021) South Korea	Randomized controlled trial To investigate the effect of walking training with a weight support feedback cane on chronic stroke patients’ lower limb muscle activity and gait ability.	Gait Assistive Devices
Kelley et al. [41] (2013) United States of America	Blinded randomized controlled trial To compare the effectiveness of robotic-assisted body weight-supported treadmill training using the Lokomat^®^ for over-ground gait training in adults with chronic stroke.	Robot-assisted gait training
Kim and Lee [33] (2013) South Korea	Randomized controlled trial To compare the effects of action observation and motor imagery training on recovery from chronic stroke.	Action observation training and motor imagery training
Kim et al. [34] (2017) South Korea	Randomized controlled trial To examine the effect of progressive backward body weight-supported treadmill training on gait in chronic stroke patients with hemiplegic gait.	Backwards walking
Lee et al. [35] (2017) South Korea	Randomized controlled trial To investigate the effects of a wearable tubing assistive walking device on gait parameters (gait speed, cadence, step length, and stride length on affected and less affected sides) in patients with stroke.	Gait Assistive Devices
Moon and Kim [36] (2017) South Korea	Randomized controlled trial To investigate the effects of the newly developed Spine Balance three-dimensional (3D) system on chronic stroke patients’ trunk strength and gait abilities.	3D Spine Balance System
Munari et al. [42] (2020) Italy	Randomized controlled trial To compare the effects of backward treadmill training versus standard forward treadmill training on motor impairment in patients with chronic stroke receiving botulinum toxin type A Therapy.	Backwards walking
Mustafaoglu et al. [43] (2020) Turkey	Randomized controlled trial To investigate the effects of robot-assisted gait training on mobility, activities of daily living, and quality of life in stroke Rehabilitation.	Robot-assisted gait training
Saleh et al. [44] (2019) Egypt	Randomized controlled trial To compare the effect of aquatic versus land motor dual-task training on chronic stroke patients’ balance and gait.	Dual-task training
Timmermans et al. [45] (2021) The Netherlands	Randomized controlled trial To compare the efficacy of two walking-adaptability interventions: a novel treadmill-based C-Mill therapy and the standard overground FALLS program.	Augmented reality-based rehabilitation
Yang [46] (2018) Taiwan	Randomized controlled trial To evaluate the effects of applying NMES over ankle dorsiflexion or plantar flexors on ankle control during walking and gait performance in chronic stroke patients.	Functional electrical stimulation
Yu et al. [47] (2020) China	Randomized controlled trial To examine the effects of body weight support Tai Chi training on balance control and walking function in stroke survivors with hemiplegia.	Tai Chi

## Data Availability

The data presented in this study are available on request from the first author.

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
