# Peer review of "Current Trends in Gait Rehabilitation for Stroke Survivors: A Scoping Review of Randomized Controlled Trials"

_jcm, 2024, doi:10.3390/jcm13051358_

Round 1
Reviewer 1 Report
Comments and Suggestions for Authors
I have some questions, which may help delve deeper into the content of the review and prompt critical analysis of the findings and implications.
1-How does the scoping review contribute to understanding the evolving landscape of gait training for stroke patients?
2-What are the main trends identified in the review that go beyond conventional rehabilitation exercises for stroke survivors?
3-How do the novel interventions identified in the review improve patient engagement and promote gait recovery?
4-What are some examples of traditional gait rehabilitation approaches with slight modifications mentioned in the review?
5-How does the integration of complementary interventions alongside conventional exercises enhance treatment outcomes for stroke survivors?
6-What are some innovative therapeutic options mentioned in the review that tap into survivors' physical and cognitive abilities?
7- How do these innovative therapies promote neural plasticity and support functional recovery in stroke survivors?
8-What are the potential benefits and limitations of incorporating modern technology, such as robotic-assisted therapy, in gait training for stroke rehabilitation?
9- What are the barriers to the widespread adoption of robotic-assisted gait training in current rehabilitation practices?
10- What further research and evaluation are needed to assess the effectiveness, cost-effectiveness, and long-term impact of robotic-assisted gait training in stroke rehabilitation?
Comments on the Quality of English LanguageThis article is intriguing and has the potential for publication after undergoing significant revisions.
Reviewer 2 Report
Comments and Suggestions for Authors
I had the opportunity to review this interesting scoping review on applications of gait rehabilitation in post-stroke rehabilitation. The manuscript is certainly interesting, but the study has significant limitations. Firstly, I have reservations about the search strategy used, which I’ve detailed further below. Moreover, a key objective of scoping reviews is to synthesize studies to identify trends and research gaps. However, this paper focuses more on individual papers, offering brief summaries in the results section akin to a narrative literature review. This approach doesn’t provide much beyond what is already available in the individual studies themselves. I have further explained my concerns below.
a. Some important databases, such as CENTRAL and Embase, were not searched, which is a major limitation.
b. Please include the full-search strategy at least for one database in the methods section.
c. Why did you limit the search to the articles published between 2013 and 2023?
d. The authors have simply summarized all of the articles included in their study, and the current paper doesn’t help identify trends in gait rehabilitation and research gaps in the field.
Reviewer 3 Report
Comments and Suggestions for Authors
Teodoro et al. have done a good review article on gait rehabilitation for stroke survivors. The methods are well explained. In my opinion, the article can be accepted after minor corrections.
1. Please employ a suitable image for each part of the result.
2. In the discussion section, provide a comprehensive table and figure of all the items stated in the results section and compare them.
3. Providing graphical abstract can be useful for this article
Round 2
Reviewer 2 Report
Comments and Suggestions for Authors
Thank you very much for inviting me to review this manuscript again, and I would like to thank the authors for their response. The comments have not been sufficiently addressed, and the manuscript still doesn’t fulfill the criteria for a robust scoping review.
Author Response
Thank you very much for your feedback. We appreciate the opportunity to revise our manuscript based on your previous suggestions. However, we respectfully disagree with your opinion regarding the adequacy of our revisions and the fulfillment of the criteria for a robust scoping review.
In the initial round of revision, we carefully addressed all of the suggestions provided and provided justifications for all methodological decisions made. These adjustments have strengthened the clarity and rigor of our manuscript.
We appreciate your time and thorough evaluation of our manuscript.